# Cooperation Between *Rhodococcus qinshengii* and *Rhodococcus erythropolis* for Carbendazim Degradation

**DOI:** 10.3390/microorganisms13010040

**Published:** 2024-12-29

**Authors:** Roosivelt Solano-Rodríguez, Fortunata Santoyo-Tepole, Mario Figueroa, Voleta Larios-Serrato, Nora Ruiz-Ordaz, Abigail Pérez-Valdespino, Everardo Curiel-Quesada

**Affiliations:** 1Departamento de Bioquímica, Escuela Nacional de Ciencias Biológicas, Instituto Politécnico Nacional, Prolongación de Carpio y Plan de Ayala S/N, Col. Santo Tomás, Mexico City 11340, Mexico; roosivelt.solano@gmail.com (R.S.-R.); viosdatafactory@gmail.com (V.L.-S.); 2Departamento de Investigación, Escuela Nacional de Ciencias Biológicas, Instituto Politécnico Nacional, Prolongación de Carpio y Plan de Ayala S/N, Col. Santo Tomás, Mexico City 11340, Mexico; fsantoyo@ipn.mx; 3Departamento de Farmacia, Facultad de Química, Universidad Nacional Autónoma de México, Mexico City 04510, Mexico; mafiguer@unam.mx; 4Departamento de Ingeniería Bioquímica, Escuela Nacional de Ciencias Biológicas, Instituto Politécnico Nacional, Av. Wilfrido Massieu, Unidad Adolfo López Mateos, Mexico City 07738, Mexico; noraruizordaz@yahoo.com.mx

**Keywords:** carbendazim, biodegradation, *Rhodococcus* spp., microbial cooperation

## Abstract

Carbendazim (CBZ) is a fungicide widely used on different crops, including soybeans, cereals, cotton, tobacco, peanuts, and sugar beet. Excessive use of this xenobiotic causes environmental deterioration and affects human health. Microbial metabolism is one of the most efficient ways of carbendazim elimination. In this work, *Rhodococcus qingshengii* RC1 and *Rhodococcus erythropolis* RC9 were isolated from a bacterial community growing in a biofilm reactor acclimated with microbiota from carbendazim-contaminated soil. Sequencing analysis of genomes of both strains revealed the presence of *cbm*A, the gene coding for the enzyme that hydrolyses carbendazim to produce 2-aminobenzimidazole (2-AB). The alternative gene for the first catabolic step (*mhe*I) was detected by PCR in strain RC9 but not in RC1. Metabolomic analysis by HPLC and LC-MS showed that both strains have the ability to metabolize carbendazim. *R. qingshengii* RC1 converts carbendazim to 2-AB, the first degradation intermediary, while *R. erythropolis* RC9 metabolizes the fungicide to its mineralization, probably because *R. qingshengii* RC1 lacks the *hdx* gene coding for 2-AB hydroxylase. HRESIMS-MS/MS results indicate that *R. erythropolis* RC9 metabolizes carbendazim by cleavage of the benzene ring and subsequent formation of 5-formyl-2-hydroxy-4,5-dihydro-1H-imidazole-4-carboxylic acid (X2 C_5_H_6_N_2_O_4_). The presence of carbendazim metabolites in culture supernatants of strains RC9 and RC1 suggests that both strains contribute to the efficient degradation of carbendazim in nature.

## 1. Introduction

Pesticides are widely used to help farmers protect crops from weeds and diseases, thus raising food productivity; however, their continued use leads to water and soil pollution, with well-documented health risks. Microorganisms are capable of degrading virtually any organic compound that is introduced to the soil environment [1]. Bacterial communities usually degrade pollutants more efficiently than individual strains since they act in a synergistic manner by optimizing nutritional and metabolic resources, which ensures their development and survival [2]. Members of the microbial community act in symbiosis via an interactive exchange of metabolic products. Degrading bacteria release metabolic intermediaries that may be harnessed by non-degraders to survive [3]. The evolution of bacterial communities requires adaptive functions and cooperation among their members and, in some instances, implies the loss or gain of genes in order to adapt to a particular environment [4]. This way, members conserving the catabolic genes act as benefactors in the development of the community [5].

Carbendazim (methyl N-(1-methylbenzimidazol-2-yl) carbamate) is a pesticide used to fight fungal diseases in crops [6,7]. It acts by inhibiting the polymerization of β-tubulin, hence interfering with microtubule formation. It is widely applied in different countries, and its use causes contamination of water and food, impairing soil sustainability and causing significant alterations in embryonic, reproductive, and developmental activities in animals and humans. Constant application of carbendazim in crops causes alterations in the stability of microbiome soil due to the enrichment of strains adapted to the fungicide [7,8,9]. Microbial degradation of carbendazim is one of the most efficient ways of eliminating this compound. Bacterial species capable of metabolizing carbendazim include *Nocardiodes* sp., which metabolizes CBZ at an average degradation rate of 0.52 mg L^−1^ h^−1^ [10]; *Rhodococcus* sp. D-1, 1.66 mg L^−1^ h^−1^; *Rhodococcus erythropolis* djl-11, 13.89 mg L^−1^ h^−1^; *R. erythropolis* JAS13, 1.56 mg L^−1^ h^−1^ [11,12,13,14]; and *Mycobacterium* sp., 0.63 mg L^−1^ h^−1^ [15].

The first step in the pathway to CBZ degradation involves the esterase MheI, which hydrolyzes carbendazim to 2-amino benzimidazole [16]. This step is also catalyzed by amidase CbmA, an alternative hydrolase more recently described [17]. The 2-amino benzimidazole (2-AB) is then transformed to 2-hydroxy benzimidazole (2-HB) [13,16], and the next step could be the opening of the imidazole ring to yield 1,2-diamino benzene, followed by catechol and its complete mineralization to CO_2_. Long et al. proposed another possible route for CBZ degradation that involves the sequence 2-AB → 2-HB → 2,6,7-HBM → X1 (C_7_H_8_N_2_O_5_) → X2 C_5_H_6_N_2_O_4_) [18,19].

The objective of this study was to characterize the genes and intermediaries involved in CBZ metabolism in order to establish the degradation pathway in two *Rhodococcus* spp. strains isolated from a microbial community recovered from agricultural soil and water in Mexico.

## 2. Materials and Methods

### 2.1. Bacterial Strains

Carbendazim-degrading *Rhodococcus* spp. strains were isolated from a carbendazim-fed biofilm reactor acclimated with bacteria from soil and water from hydroponic lettuce crops in Xochimilco, Mexico City, where the fungicide is known to be applied. Strains were initially identified by PCR amplification and sequencing of 16S rDNA as *Rhodococcus* spp. (RC1 and RC9) [20]. These turned out to be the only cultivable degraders of the bacterial community. The strains were preserved in Luria broth (LB) supplemented with 20% glycerol (J.T. Baker, Alcobendas, Spain) and stored at −70 °C.

### 2.2. HPLC and LC-MS Metabolic Analysis of Carbendazim Degrader Profiles

For the HPLC analysis, the putative carbendazim metabolizers were grown overnight in LB medium, pH 7.0, at 32 °C. Cultures were centrifuged, and the pellets were washed twice with mineral salt solution (MS) [21]. After washing, cells were inoculated into 250 mL Erlenmeyer flasks containing 100 mL of MM (MS supplemented with 30 ppm carbendazim and 1.25 g/L ammonium sulfate). Cultures were adjusted to OD600 of 0.2, incubated with shaking at 32 °C at 200 rpm, and sampled over 8 h. Every hour for 8 h, samples were centrifuged at 18,000× *g* for 1 min. Supernatants were adjusted to pH 4.0, and each sample was analyzed by a high-performance liquid chromatography system (HPLC; Infinity 1260, Agilent, Santa Clara, CA, USA) equipped with a diode array detector and a Zorbax^®^ SBC-18 column of 150 × 4.6 mm, 5 µm particle size (Agilent Technologies, Santa Clara, CA, USA). The mobile phase was ACN:H_2_O (28:72) at an isocratic flow rate of 1 mL/min and 281 nm. Carbendazim (Chem Service, Chester, PA, USA), 2-AB (Chem Service, Chester, PA, USA), and 2-HB (Merck KGaA, Darmstadt, Germany) were also analyzed as standards; all experiments were run three times.

For LC-MS analysis, strains were cultured in LB medium, washed with MS, and inoculated into flasks containing 100 mL of MM to an initial OD600 of 0.2. Cultures were incubated at 32 °C with shaking at 200 rpm. Samples were taken every two hours for 8 h and centrifuged at 18,000× *g* for 3 min. Pellets were extracted consecutively with solvents of different polarities: EtOAc, acetone, methanol, and CH_2_Cl_2_. The organic extracts were dried under a vacuum and stored until use. Culture supernatants were filtered through a 0.2 µm filter and mixed with Amberlite™ XAD (Merck KGaA, Darmstadt, Germany) 16 at a concentration of 3 g/100 mL. Mixtures were incubated overnight with shaking at 120 rpm. After this time, samples were filtered and resins were extracted successively with EtOAc, acetone, MeOH, and CH_2_Cl_2_. The filtrates were extracted with EtOAc and CH_2_Cl_2_. Extracts were treated as before. Dried samples were dissolved in dioxane/MeOH (50:50) and analyzed in an Acquity ultra-performance liquid chromatograph (UPLC; Waters, Milford, MA, USA) coupled to a mass spectrometer SQD2 (Waters) equipped with an electrospray ionization source (ESI) in both positive and negative ion modes using a Kinetex XB C18 column (1.7 μm, 100 Å, 100 × 2.1 mm; Phenomenex, Torrance, CA, USA) and ACN:H_2_O (5:95) as a mobile phase. Finally, EtOAc cell extracts were analyzed by UPLC coupled to a high-resolution mass spectrometer (Q Exactive™ Plus Hybrid Quadrupole-Orbitrap™, Thermo Fisher Scientific, Waltham, MA, USA) tandem MS-MS equipped with an ESI source (HRESIMS-MS/MS). Analyses were performed in positive mode. A BEH C18 column (1.7 μm, 130 Å, 50 × 2.1 mm; Waters) and a binary mixture of ACN and H_2_O in a linear elution gradient with an initial composition of 5% ACN up to 100% in 8 min were used, maintaining the isocratic composition for 1.5 min and returning to initial conditions in 0.5 min, with a flow rate of 0.3 mL/min.

### 2.3. Genomic DNA Extraction

Genomic DNA was extracted from *Rhodococcus* spp. strains using DNeasy PowerLyzer PowerSoil Kit, QIAGEN, CA, USA. DNA concentration and purity were determined using a Nanodrop 2000 spectrophotometer (Thermo Fisher Scientific). DNA integrity was assessed by electrophoresis on a 0.8% agarose gel.

### 2.4. Detection of the mheI Gene

Detection of the *mhe*I gene was performed by endpoint PCR using total DNA and primers *mhe*I_FW (5′-GACGCTCGAAACGCACATC-3′) and *mhe*I_RV (5′-ACGGGAACTTGACGACGACCTG-3′), designed in this study. PCR reactions were performed with 2X PCR Master Mix (NEB, Ipswich, MA, USA).

### 2.5. Genome Sequencing and Localization of Genes Involved in Carbendazim Degradation

DNA was sequenced using NEB Next Ultra II Library Prep Kit for Illumina (NEB, Ipswich, MA, USA). Adapters were ligated to each sample for library construction. Libraries were pooled in equimolar concentrations for multiplexed sequencing on the Illumina NovaSeq platform with 2 × 150 bp run parameters. FastQC v0.11. [22] was used to check the quality of sequence libraries. The Trimmomatic program v0.39 [23] was used to trim and filter raw paired readings. Reads were de novo-assembled using Spades v3.13.0 [24], and quality control of assemblies was evaluated with QUAST v5.2 [25]. Contigs were organized using Mauve software v2.4.0 [26]. Assembled genomes were annotated using RAST v2.0 [27]. The circular genome maps were generated with Proksee [28]. Finally, the search for the genes described by Long et al. [19] and their participation in carbendazim degradation was performed using BLASTN v2.16.0 and BLASTP v2.16.0 [29,30,31].

### 2.6. Phylogenomic Analysis

The bacterial strains were identified using the Virtual Analysis Method for the Phylogenomic fingeRprint Estimation (VAMPhyRE) program [32]. Genome similarity was estimated by pairwise genome comparison using virtual genomic fingerprints (VGFs). The calculated distance matrix of the VGFs was used for phylogenomic tree reconstruction using the program MEGA v11 [33] and the Neighbor-joining algorithm [34]. Phylogenomic relationships between *Rhodococcus* strains were established through whole genome comparisons by in silico DNA–DNA hybridization (*is*DDH) using the online program GGDC v3.0 [35]. To support identification, an average nucleotide identity (ANI) [36] analysis was carried out using an ANI calculator. NCBI reference genomes were used to establish sequence comparisons.

## 3. Results

### 3.1. Preliminary Screening for the mheI Gene

As a preliminary characterization stage, we searched for the gene coding for the first enzyme acting in carbendazim degradation (*mhe*I) in strains RC1 and RC9. The *mhe*I gene was found only in *Rhodococcus* RC9, which revealed a genetic difference between the strains.

### 3.2. Genetic Identification of Carbendazim-Degrading Strains

HPLC analysis of the initial degradation of individual isolates of the microbial community showed that only two strains of the community were capable of metabolizing carbendazim. The 16S rDNA sequence analysis identified both strains as *Rhodococcus* sp. The phylogenetic reconstruction tree showed that strains RC1 and RC9 grouped with *R. qingshengii* and *R. erythropolis*, respectively (Figure 1). To reinforce the phylogenetic reconstruction, the genomes of RC1 and RC9 strains were compared with the reference genomes through *is*DDH and ANI. The values of the ANI and *is*DDH between strains RC1, RC9, and their related species were all higher than 95% and 70%, respectively, which are significantly above the thresholds for species delineation. The complete genome sequences of RC1 and RC9 are available in GenBank under the accession numbers JAVIJO000000000 and JAVISC000000000.

### 3.3. Kinetic Analysis of the Carbendazim Degradation Process

Degradation kinetic analysis suggested that the *Rhodococcus* spp. strains were able to use carbendazim as a sole source of carbon. However, HPLC analysis revealed that *R. qingshengii* RC1 metabolized carbendazim only until the formation of the first intermediary 2-AB. By contrast, *R. erythropolis* RC9 degraded the carbendazim to 2-AB and then to 2-HB. These metabolites disappeared on further incubation, and carbendazim was completely depleted in around eight hours. These results suggest that strain RC9 completely mineralized the fungicide (Figure 2). Accordingly, we decided to proceed with the metabolomic characterization of RC9, while no further work was conducted on strain RC1.

### 3.4. Metabolic Profiling of Carbendazim Degradation Intermediaries by R. erythropolis RC9

In order to establish the complete degradation route, the metabolite profiling of *R. erythropolis* RC9 was examined by LC-MS. As found in the HPLC analysis, the UPLC-MS data revealed the presence of 2-AB and 2-HB (Figure 3). To get further information, the intracellular content of RC9 (EtOAc extract) was analyzed by HRESIMS-MS/MS and showed the presence of a molecular ion peak *m*/*z* 159.0395 [M+H]^+^, which corresponds to the molecular formula C_5_H_6_N_2_O_4_ (calc. for C_5_H_7_N_2_O_4_ 159.0400, Δ = −3.4 ppm, index of hydrogen deficiency of 4; retention time = 3.3 min) (Figure 3). This compound corresponds to the carbendazim degradation product X2 suggested by Long et al. [19]. Interestingly, another compound with the same molecular formula was observed in the HRESIMS-MS/MS analysis of the extract but with a different retention time (2.5 min) (Appendix A). In the time-course analysis, after 8 h of carbendazim addition, compound X2 (3.3 min) increases to more than two times its concentration; therefore, this molecule probably derives from carbendazim metabolism, and the molecule at 2.5 min could correspond to the dihydroorotic acid, an intermediary of pyrimidine biosynthesis [37,38].

### 3.5. Genetic Background Related to Carbendazim Degradation

The genomes of *Rhodococcus* RC1 and RC9 were sequenced and assembled to identify the genes responsible for carbendazim degradation. *R. qingshengii* RC1 has a circular 6.82 Mb chromosome with a GC content of 62.4% and a circular plasmid 305.266 Kb long. RAST tool annotation showed 6933 coding sequences (CDSs). Among these CDSs, we identified the *cbm*A gene responsible for the first chemical modification of CBZ to generate 2-AB. However, the *hdx* gene that converts 2-AB to 2-HB was not present; consequently, this strain exhibited partial degradation of CBZ. The rest of the genes involved in carbendazim degradation in the two pathways, according to Long et al. [19], were scattered around the chromosome (Figure 4).

The genome of *R. erythropolis* RC9 comprises a circular 6.7 Mb chromosome with an average GC content of 62.3% and one linear plasmid of 333.047 Kb. Genome annotation identified 6751 CDSs, which included genes *mhe*I and *cbm*A, responsible for the conversion of carbendazim to 2-AB. *mhe*I is located in the plasmid (Figure 4). Genes *hdx* and *edo*A, engaged in the pathway II 2-AB → 2-HB → 2,6,7-HBM → X1 (C_7_H_8_N_2_O_5_) →X2 (C_5_H_6_N_2_O_4_), were also found. The genes involved in CBZ degradation in pathway I are also present in this genome (Appendix A). It is important to highlight the presence of X2, an intermediary detected in this work (Figure 3).

## 4. Discussion

Carbendazim is a fungicide that causes alterations in human health and ecosystems. This xenobiotic is used in hydroponic lettuce crops in Xochimilco, Mexico City, where samples were taken to isolate a degrader community. The acclimated microbial consortium was capable of removing CBZ in a horizontal tubular biofilm reactor [39]. Among all cultivable bacteria isolated from the biofilm of the carbendazim-fed reactor, two *Rhodococcus* strains (*R. qingshengii* RC1 and *R. erythropolis* RC9) were able to grow in MM.

The degradation kinetics experiments revealed that strain RC9 consumed CBZ faster than RC1, clearing 80% of the carbendazim after 2 h, while RC1 metabolized the same percentage of the compound after 5 h. HPLC analyses revealed that both strains degrade CBZ by first forming 2-AB, which accumulates in strain RC1 supernatants, while it is transformed to 2-HB in strain RC9. In the last strain, both intermediaries disappear from culture supernatants after the first 8 h. The formation of these two compounds during CBZ metabolism has been reported in other bacterial strains [15]. RC9 metabolizes carbendazim (3.75 mg L^−1^ h^−1^ average degradation rate) more quickly compared to other reported strains [10,11,14,15], which suggests that this strain has a great potential to be used in bioremediation.

The *mhe*I gene codes for MheI, a hydrolase that acts at the first stage of carbendazim degradation [16]. Since strains RC1 and RC9 showed the ability to form 2-AB from CBZ, we searched for the presence of the *mhe*I gene by endpoint PCR. Polymerase chain reaction results showed the presence of the *mhe*I gene in strain RC9, but no amplicon was observed in strain RC1, which suggested that an enzyme other than MheI acts in the first stage of CBZ degradation in this strain. Whole genome analysis revealed the presence of *mhe*I in a linear plasmid in RC9 and the *cbm*A gene in the chromosomes of strains RC1 and RC9. This gene, originally described in *R. jialingiae* djl-6-2 [17], codes for CbmA, an alternative enzyme involved in the formation of 2-AB from CBZ. The *cbm*A gene is highly prevalent in the genus *Rhodococcus*, where it is extremely conserved, resulting from its vertical transmission [17], in contrast to *mhe*I, whose plasmid location explains its presence in several bacterial genera. The concurrence in RC9 of the two hydrolases involved in the initial step of carbendazim degradation and the fact that the MheI enzyme has a higher catalytic efficiency than CbmA could explain the rapid utilization of the fungicide by this strain. This concurrence could result from a high selective pressure of the fungicide, as suggested by Zhang et al. [17].

The RC1 genome encodes the fundamental enzymes required to mineralize CBZ, except for the hydroxylase Hdx that transforms 2-AB to 2-HB, which explains the accumulation of 2-AB in the supernatants of this strain. Unlike RC1, the RC9 strain has all the genes required to metabolize carbendazim; this is interesting, as RC1 and RC9 come from the same bacterial community. It is known that bacterial communities in environments contaminated with xenobiotics work in coordination to eliminate these compounds. Although not all members of a community metabolize xenobiotics, some strains have the ability to use their intermediaries, including toxic compounds, to assimilate or convert them into products that do not put the bacterial community at risk [2,3]. Since 2-AB can be recovered from *R. qinshengii* RC1 supernatants, strain RC9 could capture this intermediary for its complete degradation, suggesting that in situ, both strains could contribute to degrade CBZ more efficiently. In the same way, strain RC1 could benefit from the release of 2-HB by RC9 to complete its mineralization.

In order to outline the catabolic route followed by strain C9, we searched for the molecules formed during the degradation of CBZ. Considering that *R. qingshengii* RC1 is unable to further metabolize 2-AB, mass spectrometry analysis was not used for this strain. As mentioned before, the degradation of CBZ by *R. erythropolis* RC9 in MM led to the formation of 2-HB, as seen by HPLC. HRESIMS-MS/MS analyses of cellular extracts allowed us to detect 5-formyl-2-hydroxy-4,5-dihydroxy-1H-imidazole-4-carboxylic acid, which corresponds to compound X2 proposed by Long et al., 2021 [19]. No clear-cut evidence of the formation of any other intermediary was obtained in our analysis, probably due to their rapid catabolism. These data suggest that strain RC9 degrades carbendazim through pathway II, which includes the formation of 2,6,7-HBM followed by its conversion to X2 by the action of *edo*A-encoded extradiol dioxygenase.

*Rhodococcus* is able to degrade pollutants, including carbendazim [40]. Owing to its high tolerance to environmental stresses, such as extreme temperatures, salinization, toxic metals, desiccation, high/low pH, reactive oxygen species, and radiation exposure [41,42,43], it is considered a bio-tool. Further research will be necessary to establish the potential of using *R. erythropolis* RC9 for decontamination purposes.

## 5. Conclusions

Genomic and metabolomic analyses support the ability of *Rhodococcus* strains RC1 and RC9 to completely or partially mineralize CBZ and their cooperative interaction to degrade the fungicide. The results suggest that *R. erythropolis* C9 degrades CBZ via pathway II, proposed by Long et al., 2021 [19].

## Figures and Tables

**Figure 1 microorganisms-13-00040-f001:**
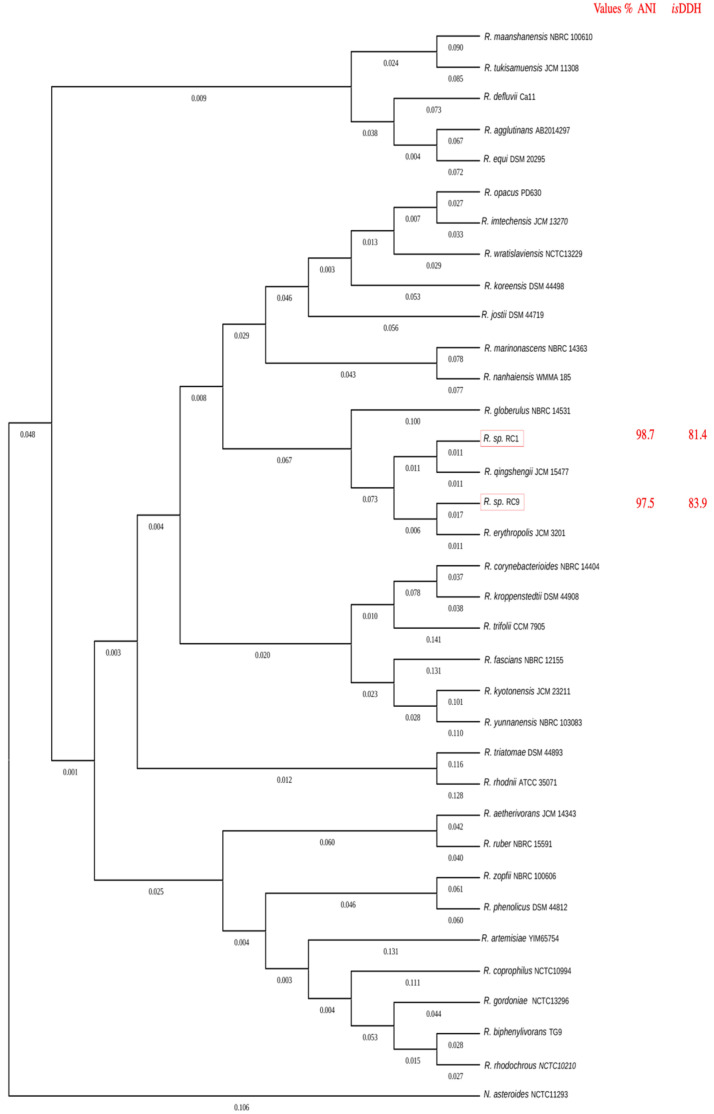
Phylogenomic analysis of whole *Rhodococcus* spp. genome sequences using VAMPhyRE. Numbers on branches indicate genetic distances. Red boxes highlight the genomes sequenced in this study. Corresponding ANI and *is*DDH values are also shown.

**Figure 2 microorganisms-13-00040-f002:**
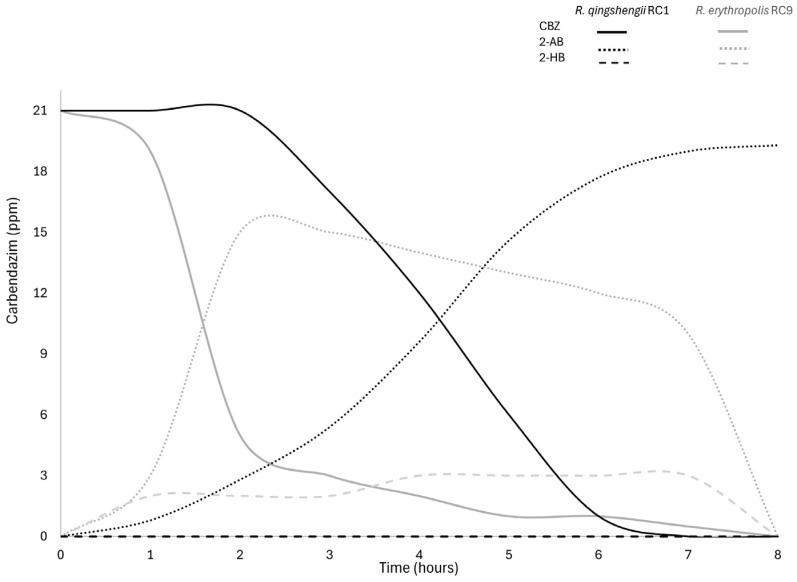
HPLC analysis of carbendazim biodegradation by *R. qingshengii* RC1 and *R. erythropolis* RC9.

**Figure 3 microorganisms-13-00040-f003:**
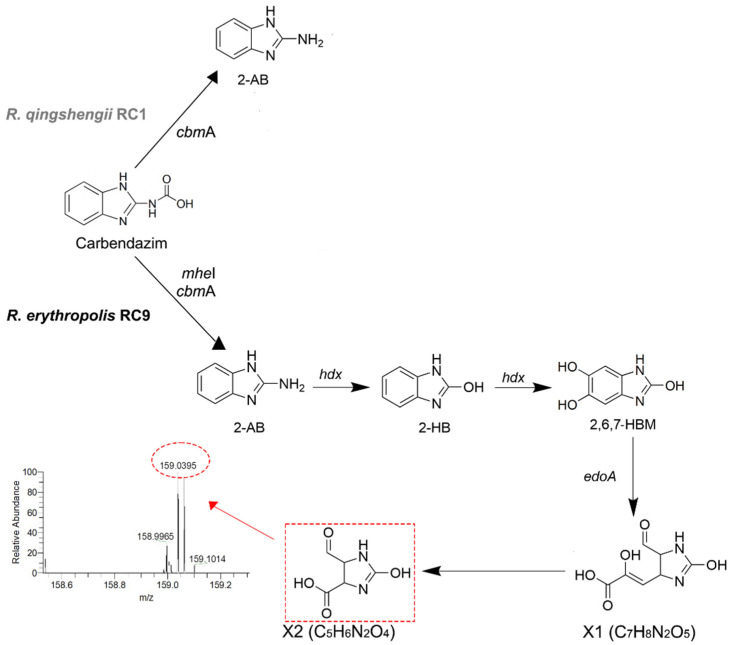
Proposed routes of carbendazim degradation by *R. qingshengii* RC1 and *R. erythropolis* RC9. The metabolite in the dotted red box was identified by HRESIMS-MS/MS (dotted red circle) in the intracellular extract of RC9.

**Figure 4 microorganisms-13-00040-f004:**
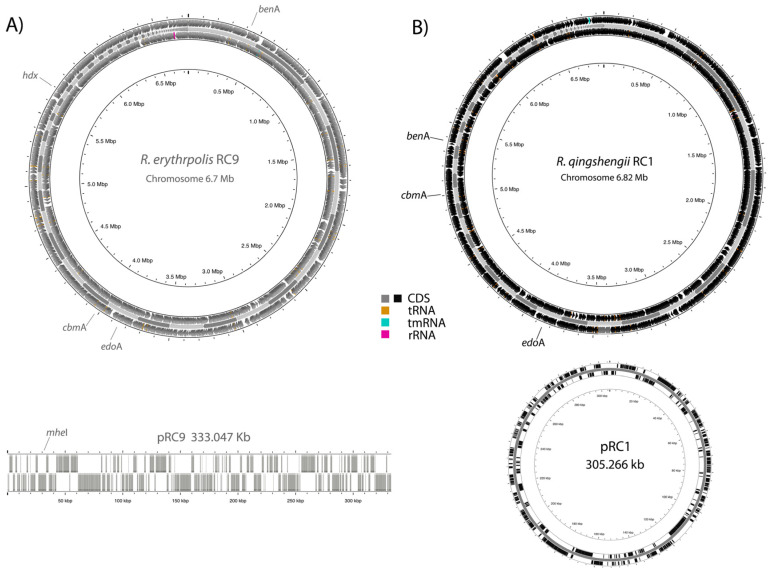
Maps of the complete chromosomes and plasmids of *Rhodococcus* strains; (**A**) RC9 and (**B**) RC1. Genes involved in CBZ degradation, proposed by Long et al. [19], are shown.

## Data Availability

The original findings of the research investigation are included in the article/Appendix A; further inquiries can be directed to the corresponding authors.

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
