# Peer review of "Cooperation Between Rhodococcus qinshengii and Rhodococcus erythropolis for Carbendazim Degradation"

_microorganisms, 2024, doi:10.3390/microorganisms13010040_

Round 1
Reviewer 1 Report
Comments and Suggestions for Authors
This manuscript investigated the isolation and characterization of two carbendazim-degrading bacteria. The possible mechanisms for carbendazim-degrading were also explored by genome sequencing analysis and metabolomic analysis of strains. In my opinion, the work is well designed and presented. The data are sufficient to support the conclusions. However, minor revisions are required to improve the quality of this manuscript.
It is difficult to draw conclusions about the relationship between the two strains. Is the title ‘Association between Rhodococcus qinshengii and Rhodococcus erythropolis in carbendazim degradation’ suitable for this paper?
In the present study, 30 ppm carbendazim was completely degraded. However, according to the introduction, investigations in microbial deduction of carbendazim have been performed. It will be helpful for the readers to better understand this work by comparing the carbendazim degradation efficiency of R. qingshengii RC1 and R. eryhtropolis RC9 with the reported strains.
Moreover, as to the carbendazim degradation experiment, how many replicates were set for each treatment?
Specific points:
Line 50, please provide the mechanism of carbendazim toward fungal diseases.
Line 51-52, please briefly summarize the ecological hazards of nontarget organisms and soil.
Line 63, X2 C5H6N2O2) [18, 19] should be X2 (C5H6N2O2) [18, 19].
Line 65, Rhodococcus should be italic.
Line 179-196, Fig. 3 depicted the proposed carbendazim degradation route by R. qingshengii RC1 and R. eryhtropolis RC9. However, the profile of metabolites of RC1 is not mentioned in the text.
Author Response
Dear Revisor,
After carefully reviewing our article proposal, based on the suggestions made by the reviewers, we have proceeded to submit it.
We would like to express our sincere gratitude to the reviewers for their dedication and hard work. Their observations have allowed us not only to significantly improve the manuscript but also to think about future research. Below we detail how we have responded to the reviewers' suggestions in the new version of our article proposal. We hope that the work carried out achieves final approval. We remain attentive to resolve any issue that may be necessary.
Sincerely,
The authors
Revisor 1
Comments and Suggestions for Authors/Answer
Dear reviewers, based on your comments or suggestions, we allow ourselves to respond to them.
This manuscript investigated the isolation and characterization of two carbendazim-degrading bacteria. The possible mechanisms for carbendazim-degrading were also explored by genome sequencing analysis and metabolomic analysis of strains. In my opinion, the work is well designed and presented. The data are sufficient to support the conclusions. However, minor revisions are required to improve the quality of this manuscript.
It is difficult to draw conclusions about the relationship between the two strains. Is the title ‘Association between Rhodococcus qinshengii and Rhodococcus erythropolis in carbendazim degradation’ suitable for this paper?
Considering your comments, we decided that the title “Cooperation between Rhodococcus qinshengii and Rhodococcus erythropolis for carbendazim…” is more appropriate for this investigation.
In the present study, 30 ppm carbendazim was completely degraded. However, according to the introduction, investigations in microbial deduction of carbendazim have been performed. It will be helpful for the readers to better understand this work by comparing the carbendazim degradation efficiency of R. qingshengii RC1 and R. eryhtropolis RC9 with the reported strains.
Information is included In lines 60, 61, 62, 241 and 242. Thanks you for your observation
Moreover, as to the carbendazim degradation experiment, how many replicates were set for each treatment?
We indicate this on line 99
Specific points
Line 50, please provide the mechanism of carbendazim toward fungal diseases.
We include the mechanism on line 53. Thank you.
Line 51-52, please briefly summarize the ecological hazards of nontarget organisms and soil.
We added this in lines 56 to 58
Line 63, X2 C5H6N2O2) [18, 19] should be X2 (C5H6N2O2) [18, 19].
We corrected subscripts, thank you for you assistance.
Line 65, Rhodococcus should be italic.
The correction was done
Line 179-196, Fig. 3 depicted the proposed carbendazim degradation route by R. qingshengii RC1 and R. eryhtropolis RC9. However, the profile of metabolites of RC1 is not mentioned in the text.
We included a more complete justification in lines 183, 184 and 185.

Reviewer 2 Report
Comments and Suggestions for Authors
Dear Authors,
This research introduces a novel bacterial strain, „Association between Rhodococcus qinshengii and Rhodococcus erythropolis in carbendazim degradation”, This research addresses the environmental and health risks of carbendazim (CBZ), a widely used but persistent fungicide. By identifying microbial strains (R. qingshengii RC1 and R. erythropolis RC9) capable of degrading CBZ, it offers an eco-friendly solution for bioremediation, reducing the fungicide's environmental impact and supporting sustainable agriculture.
However, I would like to make the following recommendations:
· Discuss the ecological and environmental importance of carbendazim biodegradation, emphasizing how the findings can contribute to solving real-world pollution issues.
· Include more specifics about the experimental conditions (e.g., culture medium, temperature, pH) to improve reproducibility and offer insight into factors affecting degradation efficiency.
· Expand the explanation of the kinetic differences between RC1 and RC9 strains, focusing on their practical implications for industrial or environmental bioremediation applications.
· Discuss potential applications of these findings in bioremediation, addressing challenges such as strain stability, environmental adaptability, or scalability of the process.
· Propose future research to explore additional enzymes, alternative degradation pathways, or the performance of these strains in more complex environmental systems like contaminated soils.
Congratulations to the authors.
Author Response
Dear reviewer,
After carefully reviewing our article proposal, based on the suggestions made by the reviewers, we will proceed to submit it.
We would like to express our sincere gratitude to the reviewers for their dedication and hard work. Their observations have allowed us not only to significantly improve the manuscript but also to think about future research. Below we detail how we have responded to the reviewers' suggestions in the new version of our article proposal. We hope that the work carried out achieves final approval. We remain attentive to resolve any issue that may be necessary.
Sincerely,
The authors
Revisor 2
Dear Authors,
This research introduces a novel bacterial strain, „Association between Rhodococcus qinshengii and Rhodococcus erythropolis in carbendazim degradation”, This research addresses the environmental and health risks of carbendazim (CBZ), a widely used but persistent fungicide. By identifying microbial strains (R. qingshengii RC1 and R. erythropolis RC9) capable of degrading CBZ, it offers an eco-friendly solution for bioremediation, reducing the fungicide's environmental impact and supporting sustainable agriculture.
However, I would like to make the following recommendations:
- Discuss the ecological and environmental importance of carbendazim biodegradation, emphasizing how the findings can contribute to solving real-world pollution issues. We included this information in lines 241,242, 243, 285, 286, 287 and 288. Thank you.
- Include more specifics about the experimental conditions (e.g., culture medium, temperature, pH) to improve reproducibility and offer insight into factors affecting degradation efficiency. We added some details in lines 87 and 91.
- Expand the explanation of the kinetic differences between RC1 and RC9 strains, focusing on their practical implications for industrial or environmental bioremediation applications.
This information was incorporated in lines 261 to 272 - Discuss potential applications of these findings in bioremediation, addressing challenges such as strain stability, environmental adaptability, or scalability of the process. Information regarding this point was added in lines 285, 286, 287, 288 and 289. Thank you for you assistance.
- Propose future research to explore additional enzymes, alternative degradation pathways, or the performance of these strains in more complex environmental systems like contaminated soils. We refer to that in lines 285, 286, 287, 288 and 289. Thanks for the observation.

Round 2
Reviewer 2 Report
Comments and Suggestions for Authors
The authors have implemented the required revisions, and in its current form, I support the publication of this manuscript.